# Development and Validation of New Exercises to Promote Physical Activity in Nursing Home Settings

**DOI:** 10.3390/geriatrics7050100

**Published:** 2022-09-16

**Authors:** Fanny Buckinx, Lucie Maton, Valentine Dalimier, Alexandre Mouton, Laetitia Lengelé, Jean-Yves Reginster, Olivier Bruyère

**Affiliations:** 1WHO Collaborating Center for Epidemiology of Musculoskeletal Health and Ageing, Division of Public Health, Epidemiology and Health Economics, University of Liège, 4000 Liège, Belgium; 2Research Unit for a Life-Course Perspective on Health & Education (RUCHE), Department of Sport Sciences, University of Liège, 4000 Liège, Belgium; 3Gérontopôle de Toulouse, Institut du Vieillissement, Centre Hospitalo-Universitaire de Toulouse, 31059 Toulouse, France

**Keywords:** physical exercise, muscle strength, balance, flexibility, gait

## Abstract

Background: GAMotion is a giant physical activity board game intended to improve levels of physical activity and a broader array of physical and psychological outcomes among nursing home residents. Objective: The aim of the present study is to develop and validate new balance, flexibility, muscle strength, and walking exercises to be included in GAMotion. Methods: A two-step design combining the Focus group and Delphi method was conducted among healthcare professionals divided into two independent samples of experts. The first sample was asked to develop exercises during a focus group. The second sample participated in a two-round Ranking-type Delphi method. During the first round, the participants were asked to rate the exercises developed during the focus group on a four-point Likert scale (from 1: not adapted at all to 4: very adapted). The exercises that did not reach consensus were removed (consensus established: median ≥ 3 on the Likert scale and at least 75% of experts rating the exercises as « adapted » or « very adapted »). During the second round, it was asked to rank the exercises selected at the end of the first round from most suitable to least suitable. Results: The Focus group developed nine balance, twelve flexibility, twelve strength, and nine walking exercises. Following the first round of the Delphi method, two exercises in each category did not reach a consensus and were then removed. In the second round, the remaining seven balance, ten flexibility, ten strength, and seven walking exercises were ranked by the experts, and this classification allowed us to determine the four most suitable exercises from each category to be included in the GAMotion. Conclusion: A consensus-based approach among healthcare professionals allowed us to contribute to the development of new exercises to promote physical activity in nursing homes. These validated exercises can be included in the GAMotion board game.

## 1. Introduction

Normal aging is accompanied by a deterioration in functional and locomotor capacity [1,2], accentuated by physical inactivity and a sedentary lifestyle, which affects 50% of the elderly [3]. In addition, this deterioration decreases the mobility of older adults, creating a vicious cycle of deconditioning [3], which accelerates the spiral of loss of autonomy and sarcopenia and increases the need for health care and services and, therefore, health costs [4,5]. Unfortunately, inactivity and sedentary problems are even more prevalent in nursing homes. In fact, nursing home residents spend the majority of their time inactive [6], and they walk on average 1678 ± 1621 steps per day, which is far from the recommended levels that advocate for a minimum of 3000 steps/day [7,8]. However, physical inactivity is the fourth risk factor for mortality [9]. In addition, a lack of physical activity is detrimental to older adults’ health, functional independence, and quality of life [3]. Sedentary risk factors include intrinsic factors (e.g., physical health, attitudes related to aging, financial costs, lack of motivation, enjoyment, lack of companionship, and knowledge of programs), extrinsic factors (e.g., transports, limited availability of physical activity programs and lack of information on available activities, culture, and sense of acceptance) and health-related factors (e.g., musculoskeletal disorders). On the other side, it is admitted that the implementation of physical activity interventions leads to positive effects on functional ability, cognition, or mood in older adults [10,11,12,13]. In fact, being active is associated with body composition and functional capacities [14]. In addition, physical activity seems to improve mitochondrial density and dynamics (i.e., resistance training) and is related to mitochondrial antioxidant capacity improvements (i.e., endurance training) [15]. Thus, Interventions should encourage the oldest adults to reduce sedentary time and especially target mentally passive sedentary time [16].

This is why our team developed, a few years ago, a giant physical activity board game, GAMotion, in order to promote physical activity in nursing homes. It is a medical device of class 1. As shown in Figure 1, the GAMotion measures 3.5m long and 1.5 m wide and is composed of 12 squares divided into three distinct colors corresponding to the three main components of physical activity: balance (four purple squares); muscle strength (four yellow squares); and walking or endurance (four red squares). In addition, the mat has seven squares which represent the walking path to perform the walking exercises (red squares). On each square, three levels of difficulty are represented by one, two, or three stars so that the exercises are suitable for the fitness levels of all participants. The principle of the game is similar to the traditional goose game, and the game only requires a die and a chair.

Two previously published studies support the positive effects of GAMotion on the level of physical activity and a broader array of physical and psychological outcomes [17,18]. In the first publication, we showed that nursing home residents who used the GAMotion for a 1-month period (3 times a week) significantly increased their daily number of steps and their daily energy expenditure but also their quality of life, balance, gait, and strength of the ankle. More interestingly, these improvements still persisted 2 months after stopping the GAMotion intervention [18]. In the second publication, residents included in a 1-month intervention using GAMotion displayed greater improvement in Tinetti score, Timed Up and Go test, Short Physical Performance Battery test (SPPB), knee extensor isometric strength, grip strength, symmetry of steps, three domains of the EQ-5D (i.e., mobility, self-care, usual activities), and intrinsic motivation, compared to the control group [17].

These results are promising but still have some limitations. First, a known barrier to the practice of physical activity among seniors is the lack of a variety of exercises. Indeed, older adults, even in nursing homes, need variety and innovation in their exercise programs [19,20,21]. We should acknowledge that GAMotion may seem monotonous due to the limited number of exercises offered. Second, the literature recommends flexibility training to supplement other forms of exercise to improve the functional ability of older adults [22]. To date, the flexibility category is not present in GAMotion, and this study will allow us to add this category of exercise in a new version of the giant board game.

Thus, with a view to improving the current version of GAMotion and countering the limitations mentioned above, the aim of this study was to develop four new balance, four flexibility, four muscle strength, and four walking exercises to be included in GAMotion (to replace the current exercises), to promote physical activity in nursing home settings.

## 2. Methods

### 2.1. Study Design and Participants

A two-step design combining the Focus group and Delphi method was conducted among healthcare professionals working in nursing homes or who have contact with nursing home residents. The philosophical research paradigm used to guide our design is positivism because our methods resulted from foundationalism and empiricism. In fact, we valued objectivity and proving or disproving our hypotheses [23].

The recruitment of the participants was performed in April 2021 via invitations sent by email to the directors of 807 nursing homes from the French-speaking part of Belgium as well as via announcements on social media. If they were interested in this study, healthcare professionals were invited to contact us by email, phone, or via social media.

To be included in this study, the healthcare professionals must belong to one of the following categories: (1) physiotherapists working in nursing homes, (2) occupational therapists working in nursing homes, (3) physical educator specialists in adapted physical activity, working in nursing homes, (4) nurses working in nursing homes, (5) general practitioners (GP) with patients residing in nursing homes, (6) physiotherapist students who have completed at least one internship in nursing homes. No specific exclusion criteria were defined.

The healthcare professionals were divided into two samples: the first sample participated in a focus group, while the second sample participated in the Delphi method.

### 2.2. Focus Groups

The first sample was asked to develop new exercises. To do this, a focus group lasting 2 h was organized by videoconference, via Teams media, on 25 May 2021. The focus group was conducted in several steps, validated by an expert in qualitative method: (1) welcome and introduction of each participant (age, sex, and occupation were asked to each participant), (2) presentation of the objectives of the discussion, (3) presentation of the GAMotion board game, (4) discussion about the difficulties encountered by the residents, (5) development of exercise ideas for each physical activity categories: balance, flexibility, muscle strength and walking exercises, (6) questions/answers session, (7) conclusion.

During steps 4 and 5 of the focus group, the different ideas emanating from the participants were noted on a virtual board using MURAL© software. That way, the participants could observe this virtual board via screen sharing. The ideas were organized into diagrams according to each exercise category directly.

### 2.3. Delphi Method

The second sample participated in a two-round ranking-type Delphi method, each with a questionnaire. This questionnaire included questions related to the socio-demographic characteristics of the participants: age, sex, and occupation. We followed the guidelines proposed by Kobus et al. to conduct a rigorous ranking-type Delphi [24]. It is widely used since 93% of papers published between 1984 and 2010 used this type of Delphi analysis [25]. The ranking-type Delphi method is well suited as a means for consensus-building by using a series of questionnaires to collect data from a panel of geographically dispersed participants [25].

#### 2.3.1. First Round

The first questionnaire asked participants to rate each exercise developed during the focus group on a 4-point Likert scale (1: not adapted at all; 2: not adapted; 3: adapted; 4: very adapted). In addition, a note about each exercise had to be issued by the respondents.

The link to the questionnaire, created using Google form, was sent by email on 28 May 2021, and the participants were asked to answer within two weeks. Two reminders were sent, by email, 7 and 3 days before the deadline.

At the end of the first round, the mean (±SD) and the percentage of each answer were calculated for each exercise (i.e., score on the Likert scale). The exercises that did not obtain a positive consensus were removed from the list for the second round. A positive consensus was established when the mean score was above or equal to 3 and when at least 75% of the responses were “adapted” or “very adapted”.

#### 2.3.2. Second Round

Based on the results obtained in the first round, a second questionnaire was developed, and the participants were asked to rank the selected exercises, from most suitable to least suitable in each category (i.e., muscle balance, flexibility, muscle strength, and walking exercises).

The link to the questionnaire, created using Google form, was sent by email on June 14th, 2021, and the participants were asked to answer within two weeks. Two reminders were sent by email, 7 and 3 days before the deadline.

At the end of this second round, the 4 highest ranked exercises in each category were selected to be included in GAMotion. To do this, points were awarded to the exercises based on their position in the ranking. For muscle strength and walking exercises, 10 points were awarded to the exercise placed first (the most suitable), 9 points were awarded to the exercise placed in the second position, and so on until the last exercise (least suitable), to which 1 point was awarded. For balance, 7 points were awarded to the exercise placed at the top of the classification, 6 points to the next, and so on, continuing until the last exercise considered to be the least suitable (1 point).

### 2.4. Statistical Analysis

Statistical analyses were performed using R software (version 4.0.3) and its extension R commander (version 2.7-1) software. Quantitative ordinal data were expressed as mean and standard deviation (SD), while the categorical variables were expressed as numbers and percentages.

#### 2.4.1. Focus Group

MURAL© software was used to analyze the data obtained during the focus group. Using MURAL, the participants could observe the virtual board via screen sharing. The ideas arising from the discussion were organized into diagrams according to each exercise category. That way, a list of exercises was obtained at the end of the focus group.

#### 2.4.2. Delphi Method

At the end of the first round of the ranking-type Delphi method, a consensus was calculated for each exercise. Following the second round of the Delphi method, the ranking method was used to classify the exercises, and the mean score (±SD) on the Likert scale was also calculated for each exercise.

## 3. Results

### 3.1. Participants

As shown in Figure 2, a total of 32 healthcare professionals took part in the study (i.e., eight in the focus group and 24 in the Delphi method).

Following the invitations, 22 people expressed their interest in participating in the study by email (18 physiotherapists, two occupational therapists, one special educator, and one director). The director and the special educator did not meet the selection criteria and were, therefore, excluded from the study. A total of three physiotherapists and one occupational therapist were finally selected according to their common availability. In addition, one GP, one nurse, one physical educator specializing in adapted physical activity, and one physiotherapist student expressed their interest in the focus group through social networks and were enrolled. In this way, eight healthcare professionals were enrolled in the focus group.

Healthcare professionals interested in the focus group but not included had the opportunity to participate in the Delphi method, and three physiotherapists were included. In addition, nine healthcare professionals (six physiotherapists and three occupational therapists) expressed their interest in participating in the Delphi method by email, while 12 others (three GPs, three nurses, three physical activity educators, and three physiotherapist students) expressed their interest via social media. Ultimately, 24 subjects were enrolled in the Delphi method. Note that out of the 24 healthcare professionals included in the Delphi method, only 15 (62.5%) participated in the second round of the Delphi method. The nine other subjects did not justify their abandonment.

The general characteristics of the healthcare professionals included in the two steps of the study are presented in Table 1. More than half of the samples were made up of women. Moreover, the samples included three times more physiotherapists than other professions.

### 3.2. Focus Group

First, the healthcare professionals discussed the difficulties encountered by nursing home residents in daily living regarding balance, flexibility, muscle strength, and walking.

Regarding balance, the following difficulties were suggested by the healthcare professionals: (1) difficulty maintaining balance when changing direction, (2) difficulty maintaining balance when the patient picks up an object from the ground or from above, (3) difficulty for patients who have vision or hearing problems, (4) difficulty maintaining balance on uneven or sloping ground, (5) and difficulty maintaining balance while climbing stairs.

Regarding flexibility, the following were highlighted: (1) difficulty moving the pelvis, (2) difficulty moving the shoulders, (3) difficulty picking up an object from the ground, and (4) difficulties in daily activities such as uncorking a bottle, tying shoelaces, doing one’s hair, fastening buttons/snaps, etc.

Regarding muscle strength, these difficulties were listed: (1) difficulty in carrying heavy objects, (2) weakness in the legs, and (3) generalized weakness and fatigue.

Regarding walking, experts mentioned the following: (1) no rolling of the foot, (2) no step attack with the heel, (3) residents who move with a technical aid leaning too far forward, (4) feet cross while walking, (5) the stride is too short, and (6) difficulty walking in double tasks.

Based on these highlighted difficulties, nine balance, twelve flexibility, twelve strength, and no walking exercises were created by the healthcare professionals during the focus group to overcome the difficulties encountered by the nursing home residents. These exercises are detailed in Table 2 below.

### 3.3. Delphi Method

The ranking-type Delphi method was carried out in two rounds.

#### 3.3.1. First Round

Firstly, a consensus was reached for 7/9 balance exercises (77.8%). Exercise number 8 obtained an average score of four on the Likert scale. However, exercises 5 and 9 obtained a mean score lower than three on the Likert scale and did not reach a 75% consensus. They were therefore eliminated.

Secondly, a consensus was reached for 10/12 flexibility exercises (83.3%). Although all of the balance exercises had an average score above three on the Likert scale, two of them did not reach a 75% consensus and were therefore removed (exercises 9 and 10).

Thirdly, a consensus was reached for 10/12 strength exercises (83.3%). Five strength exercises scored 100%, including three with a mean score of four on the Likert scale. Exercises 4 and 10 had a mean score below three on the Likert scale and have not reached at least a 75% consensus. Therefore, they were not selected for the second round of the Delphi method.

Finally, a consensus was reached for 7/9 walking exercises (77.8%). Two walking exercises (exercises 5 and 6) obtained an average score of four on the Likert scale, while the others obtained a mean score of three. However, exercises 1 and 7 did not reach a 75% consensus and were therefore removed.

These respondents commented on the exercises, and these remarks mainly concerned the effectiveness of the exercises, the safety, the hygiene, the adequacy or the complexity for nursing home residents, and the possible variants. For example, a walking exercise on the heels, raising the point of the feet, was proposed. A comment concerning this exercise was “probably impossible because already complex for able-bodied people”. Another example of comment concerns the stride of the dice placed on the ground: “it is a useful but complicated exercise for some residents who have difficulty bending their knees and lifting their feet off the ground”. However, as an example, a proposed exercise consisted of joining the hands and raising them as high as possible. We received the following comment “it is an exercise often used in nursing homes and very popular with residents”.

#### 3.3.2. Second Round

The ranking of the four best exercises by category (illustration and average ranking), according to health professionals, is presented in Table 3. These exercises are therefore considered scientifically validated.

## 4. Discussion

The main objective of this study was to develop and validate four balance, four flexibility, four muscle strength, and four walking exercises that can be used to promote physical activity in nursing homes, especially using the GAMotion. Following the focus group and the Delphi method, seven balance, ten flexibility, ten muscle strength, and seven walking exercises were classified in order to ultimately highlight the four most suitable exercises in each category.

For exploratory research and data collection, focus groups are considered an appropriate methodology and are increasingly being used [26]. According to Stalmeijer et al., it is most often used as a starting point for research, as is the case with our study [26]. Jorm also confirms that focus groups are a common method for researching items to be included in a questionnaire intended for the Delphi method [27]. Jandhyala’s study also found that 11.89% of Delphi studies used meetings between experts to generate the initial elements [28]. The literature recommends including six to ten individuals in a focus group [26], and our sample corresponded to this criterion since it was composed of eight healthcare professionals. Moreover, our sample was heterogeneous because the participants came from diverse backgrounds, which stimulated the discussion and also invited subjects to broaden their critical thinking [26].

Regarding the Delphi method, the results can be considered stable with panels of 20 or more people [27]. Our study included 24 subjects in the Delphi method, which seems adequate. In addition, the individuals included in a Delphi method must be subject matter experts [29], as was the case in our study. Note that the Delphi method is generally used in the case of a lack of evidence and research on the chosen topic [27], which was the case for the present study. The study of McMillan et al. suggests that the first round of the Delphi method presents statements that participants rate on a clearly defined Likert scale [29]. Thus, in our study, the participants were asked to rate the exercises using a four-point Likert scale. Indeed, we did not want to offer the possibility of selecting a “neutral” answer. This choice is also found in the literature [30,31]. Moreover, a note concerning each exercise was requested during the first round of the Delphi method because asking experts to comment on their choice is recommended in the literature [29,32]. This approach enabled exploration of the limitations and further recommendations from the expert’s point of view. Then, the number of rounds varies from one study to another. In most studies, two rounds are used [29], and this is the optimal number according to the literature) [33]. Globally, the Delphi method can stop when consensus is reached, depending on the objectives of the study. In our study, two rounds were needed. Note that all participants in round 1 were invited to participate in round 2, as is usually the case [33]. The notion of consensus is a sensitive point of the Delphi method. In a systematic review conducted among 100 studies, the most frequent definition of consensus was based on the percentage of agreement, and 75% was the median threshold for defining consensus [34]. Several studies using a four-point Likert scale [28,33,34] also established their consensus at 75%. We therefore based these works to establish the consensus in our study.

The exercises have been chosen to respond to the difficulties of older adults observed by healthcare professionals regarding walking, balance, muscle strength, and flexibility. This is consistent with the literature because functional training may be a better exercise program for older adults to improve independence in ADL [35]. Thus, older adults should be trained on specific tasks, such as chair rise or movements needed to carry out daily tasks [35], as is the case in GAMotion. In addition, there is strong evidence that the combination of muscle strengthening, balance, endurance, and flexibility exercises minimizes fall risk in older adults [12,36], and the game uses these four components of physical activity. However, the specific exercises to be prescribed in each of these categories are not defined in the literature. In fact, these exercises should be adapted to each person’s physical condition.

The strength of this study lies in the rigorous methodology used to develop and validate the exercises, but we must recognize that our study has some limitations. First of all, there is representativeness bias because not all professional categories related to nursing homes are represented. However, the professions most related to GAMotion were included. Then, a recruitment bias is present since only volunteers were included via social media and emails. An information bias is also present since data regarding the length of experience in relation to the application of physical exercises for the elderly and the academic background in that area have not been collected. In addition, there was a significant dropout rate between the two rounds of the Delphi Method (round 1 *n* = 24, round 2 *n* = 15). Unfortunately, the reasons for abandonment are not known. This non-adhesion rate can be attributed to the relatively short, expected time to respond (two weeks) or to the period in which it was launched (end of the academic year). Finally, a limitation concerns the statistical analysis because only one researcher performed the analysis, and therefore, the credibility of the results has not been tested (e.g., kappa–Cohen inter-judges).

Thanks to the results of this research, the future perspective is the creation of a new version of GAMotion, using the newly developed and validated exercises, in order to propose more variety in the exercises. Another perspective is the setting up of a randomized controlled study to establish the effectiveness of the new exercises on the level of physical activity, physical abilities, motivation, and quality of life among nursing home residents. Another perspective is to assess the effect of GAMotion on fine motor skills of nursing home residents. For now, it is difficult to compare the results of this study with those found in the scientific literature because studies do not specifically mention the exercises used in physical activity programs intended for older adults. However, we proposed a multi-component physical activity program, and numerous studies performed in nursing home settings recommend implementing combined physical activity intervention. In this sense, a meta-analysis including 12 studies concluded that a combined PA program [type: strength and balance; frequency: 2 to 3 times/week; duration: 6 months and more] would prevent falls in nursing home residents with reduced mobility [37] Moreover, a systematic review showed that practicing combined moderate-intensity PA is the best intervention to improve quality of life, autonomy, balance, and anxiety in frail older adults [38].

## 5. Conclusion

In conclusion, a consensus-based approach among healthcare professionals allowed us to contribute to the development of four new balance, four flexibility, four muscle strength, and four walking exercises to promote physical activity in nursing home settings. These validated exercises can be included in the GAMotion board game. This research will allow the clinician to vary the exercises proposed during the revalidation and physiotherapy sessions.

## Figures and Tables

**Figure 1 geriatrics-07-00100-f001:**
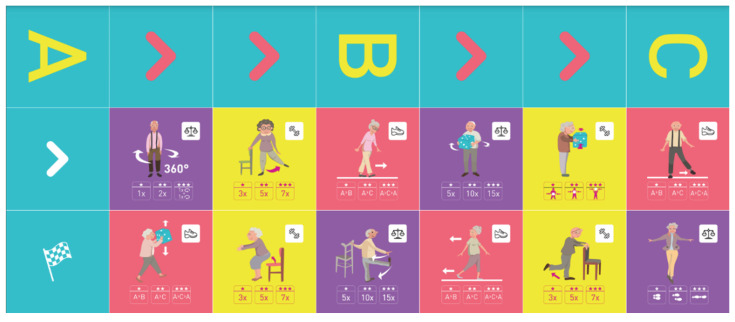
Illustration of the GAMotion board game.

**Figure 2 geriatrics-07-00100-f002:**
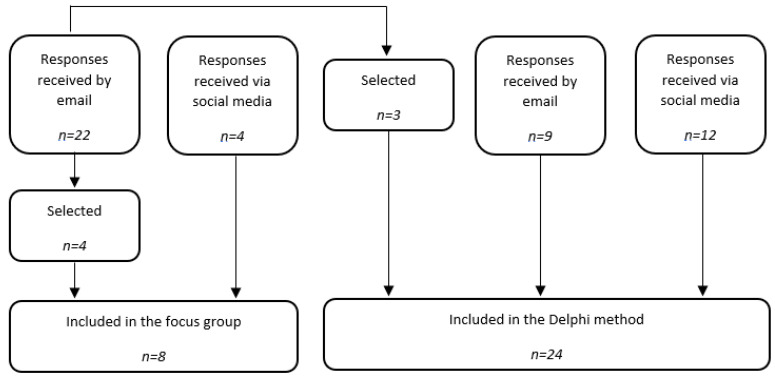
Flowchart of the recruitment of the healthcare professionals.

**Table 1 geriatrics-07-00100-t001:** General characteristics of the healthcare professionals included in the study (*n* = 32).

	Focus Group (*n* = 8)	Delphi Method (*n* = 24)
Age (years)	41.5 ± 14.6	36.7 ± 11.7
Sex (women)	5 (62.5%)	18 (75%)
Occupation		
GP	1 (12.5%)	3 (12.5%)
Nurse	1 (12.5%)	3 (12.5%)
Physiotherapist	3 (37.5%)	9 (12.5%)
Occupational Therapist	1 (12.5%)	3 (12.5%)
PA Educator Specialized in Adapted Physical Activity	1 (12.5%)	3 (12.5%)
Physiotherapist Student	1 (12.5%)	3 (12.5%)

GP = general practitioner; PA = physical activity; Continuous variables are expressed as mean and standard deviation; Categorical variables are expressed as number and percentage.

**Table 2 geriatrics-07-00100-t002:** Balance, flexibility, muscle strength and walking exercises proposed by the healthcare professionals during the focus group.

	Balance Exercises (*n* = 9)	Flexibility Exercises (*n* = 12)	Muscle Strength Exercises (*n* = 12)	Walking Exercises (*n* = 9)
1	In a standing position with parallel feet, close your eyes and maintain your balance (10 s)	Rotate the dice around the head (5 reps)	Stand leaning on a chair, stand on tiptoes 10 reps)	Walk on heels, tiptoes up (3.5 m)
2	Stand leaning on a chair, raise the knee and opposite arm (5 s on each side)	Join hands and raise them as high as possible (5 reps)	Stand leaning on a chair, bend the knees, or squats (5 reps)	Walk on tiptoes (3.5m)
3	In a standing position, turn the head from right to left and then from left to right while keeping your balance (5 reps)	Stretch out your arms in front of you with the dice in your hands and make circles (5 reps in one direction and then 5 reps in the other direction)	Stand leaning on a chair, raise the knee (5 reps on each side)	Walk with a die in your hands, moving it from left to right and then from right to left (3.5m)
4	In a standing position, tilt the head forward and then back while keeping your balance (5 reps)	While seated on a chair, self-grow as much as possible (10 s)	Starting from a standing position, feet parallel, take a step forward and bend the front knee, and then return to the starting position (5 reps on each side)	In a standing position facing the dice, step over the dice with one leg and return to the starting position (5 times with the right leg and 5 time with the left leg)
5	In a standing position, feet pelvis width, shift the weight of the body in the heels and then on the toes (5 reps)	In a standing position, turn the head (5 reps in one direction and then 5 reps in the other direction)	In a seated position on a chair, press the dice with your knees (5 reps)	Walk looking ahead (3.5 m)
6	In a standing position, facing the dice, put one foot on the dice and then bring it back to the ground (5 reps on each side)	Bring the hands behind head to touch each other (5 reps)	Stand leaning on a chair, lift straight leg back, or hip extension (5 reps on each side)	Walk on either side of a line (3.5 m)
7	In a standing position, lift one foot and touch the adjacent squares: front, side, and back) (5 reps on each side)	Bring the hands behind the back to touch each other (5 reps)	Sitting on a chair, put your hands on the armrests and push on your hands so as to slightly lift your buttocks from the chair, or dips (5 reps)	Walk by rolling the dice up and catching it (3.5 m)
8	In a standing position, facing the dice, crush the dice with one foot (5 reps on each side)	Sitting in a chair, touching the knees and then the ankles (5 reps)	With a bottle of water in each hand, lift your arms outstretched forward, or front raises (5 reps)	Walk by putting one foot in each square of the mat (3.5 m)
9	In a standing position, lift the right knee and touch it with the left hand, and vice versa (5 reps on each side)	Seated on a chair, place the left heel on the right knee and then vice versa (5 reps on each side)	With the dice in your hands, raise your arms outstretched above your head (5 reps)	Walk with a folded towel over your head without dropping it
10		Standing facing a chair, touching the seat of the chair with your foot (5 reps on each side)	Sitting on a chair, throwing the dice up and catching it (5 reps)	
11		In a seated position, rotate the ankles (5 reps in one direction and then 5 reps in the other direction)	With a bottle of water in each hand, bend the elbows to 90° and then perform an external rotation of the shoulder and then an internal rotation (5 reps)	
12		In a standing position, rotate the pelvis (5 reps in one direction and then 5 reps in the other direction)	Stand leaning on a chair, step back one foot and bend the knees, or lunge (5 reps on each side)	

**Table 3 geriatrics-07-00100-t003:** Top 4 for balance, flexibility, muscle strength and walking exercises obtained at the end of a Delphi method conducted among healthcare professionals.

	Balance Exercises		Flexibility Exercises		Muscle Strength Exercises		Walking Exercises	
Ranking	Top 4	Mean Score	Top 4	Mean Score	Top 4	Mean Score	Top 4	Mean Score
1	Exercise 1 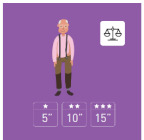	2.8 ± 1.97	Exercise 2 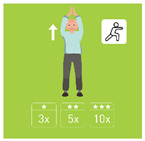	1.66 ± 1.14	Exercise 2 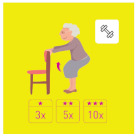	3.07 ± 1.94	Exercise 6 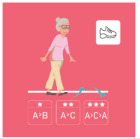	1.8 ± 2.07
2	Exercise 6 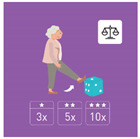	3.47 ± 1.68	Exercise 4 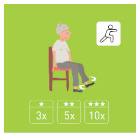	3 ± 1.41	Exercise 1 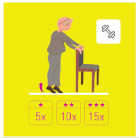	3.33 ± 3.09	Exercise 8 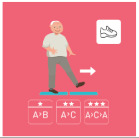	2.33 ± 1.52
3	Exercise 7 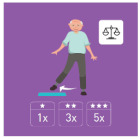	3.53 ± 1.8	Exercise 11 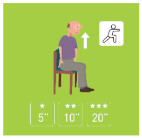	4.93 ± 3.56	Exercise 3 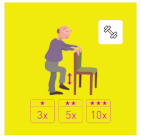	4.07 ± 2.76	Exercise 5 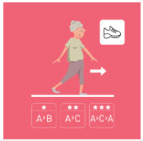	2.47 ± 1.52
4	Exercise 2 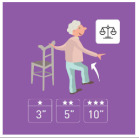	3.6 ± 2.32	Exercise 3 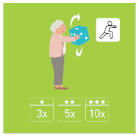	5 ± 2.94	Exercise 5 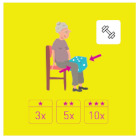	4.73 ± 2.87	Exercise 4 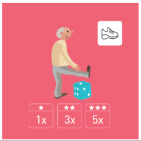	3 ± 1.54

The mean score is presented as mean ± SD and represent the mean score assigned by the healthcare professionals on the Likert scale.

## Data Availability

The data presented in this study are available on request from the corresponding author. The data are not publicly available due to ethical restrictions.

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
