# Peer review of "Development and Validation of New Exercises to Promote Physical Activity in Nursing Home Settings"

_geriatrics, 2022, doi:10.3390/geriatrics7050100_

Round 1
Reviewer 1 Report
The paper focused on exercise training for nursing home residents. The present article provides scientific support for this new approach by using a bibliometric analysis to explore important points and frontiers of research in this field. However, the comments below need to be answered before the manuscript can be considered for acceptance.
Introduction
Please make a clearer aim. What is the primary, secondary? What new will be added in the field?
Methods
Please say more about samples' recruitment.
How did you calculate samples' size?
Inclusion/exclusio criteria
Results
Sample: Any drop outs?
Discussion
PLease include strengths of the study, limitations, future research suggested.
Statement: Past studies have confirmed the effectiveness of exercise training in nursing home residents: Provide a citation for this important statement
Effects of multicomponent exercise training intervention on hemodynamic and physical function in older residents of long-term care facilities: A multicenter randomized clinical controlled trial. Journal of Bodywork and Movement Therapies 28, 231-237.
https://doi.org/10.1016/j.jbmt.2021.07.009
1. Pepera G, Krinta K, Mpea C, Antoniou V, Peristeropoulos A, Dimitriadis Z (2022). Randomized Controlled Trial of Group Exercise Intervention for Fall Risk factors Reduction in nursing home residents. Canadian Journal on Aging, 42 (1). https://doi.org/10.1017/S0714980822000265
TrTraining Improves Strength, Balance, and Gait Speed in Elderly Nursing Home Residents. Clin Interv Aging. 2020 Feb 7;15:177-184. doi: 10.2147/CIA.S234299. eCollection 2020.
Author Response
The paper focused on exercise training for nursing home residents. The present article provides scientific support for this new approach by using a bibliometric analysis to explore important points and frontiers of research in this field. However, the comments below need to be answered before the manuscript can be considered for acceptance.
Introduction
Please make a clearer aim. What is the primary, secondary? What new will be added in the field?
Authors’ response : We fully agree with the reviewer. The purpose of the study is now more clearly stated at the end of the introduction:
« Thus, with a view to improving the current version of the GAMotion and countering the limitations mentioned above, the aim of this study was to develop 4 new balance, 4 flexibility, 4 muscle strength and 4 walking exercises to be included in the GAMotion, to promote physical activity in nursing home settings. »
In addition to the previous studies related to the GAMotion, the present study brings more variety in the validated exercises and includes flexibility exercises. These elements are now described in the introduction section:
« These results are promising but still have some limitations. First, a known barrier to the practice of physical activity among seniors is the lack of variety of exercises. Indeed, older adults, even in nursing home, need variety and innovation in their exercise programs. We should acknowledge that the GAMotion may seem monotonous due to the limited number of exercises offered. Second, the literature recommend flexibility training, as supplement to other forms of exercises to improve functional ability of older adults [18]. To date, the flexibility category is not present in the GAMotion and this study will allow us to add this category of exercise in a new version of the giant boardgame. »
Methods
Please say more about samples' recruitment.
How did you calculate samples' size?
Inclusion/exclusio criteria
Authors’ response: The selection criteria of the population are described in the method section :
« To be included in this study, the healthcare professionals must belong to one of the following categories: 1) physiotherapist working in nursing home, 2) occupational therapist working in nursing home, 3) physical educator specialist in adapted physical activity, working in nursing home, 4) nurse working in nursing home, 5) general practitioner (GP) with patients residing in nursing home, 6) physiotherapist’s student having completed at least one internship in nursing home. No specific exclusion criteria were defined ».
Regarding the sample size, a calculation of statistical power was not possible since it is not an interventional study. However, we based on the litterature which recommend to include 6 to 10 peoples in a focus group and at least 20 people in a Delphi method. This important justification has been added to the discussion section :
- « For exploratory research and data collection, focus group is considered as an appropriate methodology and is increasingly being used [19] » …
- “Regarding the Delphi method, the results can be considered as stable with panels of 20 or more people [25]. Our study included 24 subjects in the Delphi method, which seems adequate.”
- « The literature recommends to include 6 to 10 peoples in a focus group [19] and our sample corresponded to this criterion since it was composed of 8 healthcare professionals. »
Results
Sample: Any drop outs?
Authors’ response : It is an important point that has been now included in the manuscript :
« In addition, there was a significant dropout rate between the two rounds of the Delphi Method (round 1 n = 24, round 2 n = 15). Unfortunately, the reasons for abandonment are not known ».
Discussion
PLease include strengths of the study, limitations, future research suggested.
Authors’ response :Strengths, limitations and perspective have been discussed in the appropriate section:
« The strength of this study lies in the rigorous methodology used to develop and validate the exercises but we must recognize that our study has some limitations. First of all, there is representativeness bias because not all professional categories related to nursing homes are represented. However, the professions most related to the GAMotion were included. Then, a recruitment bias is present since only volunteers were included via social media and emails. In addition, there was a significant dropout rate between the two rounds of the Delphi Method (round 1 n = 24, round 2 n = 15). Unfortunately, the reasons for abandonment are not known. This non-adhesion rate can be attributed to the relatively short expected time to respond (two weeks) or to the period in which it was launched (end of the academic year).
Thanks to the results of this research, future perspective is the creation of a new version of the GAMotion, using the new developed and validated exercises, in order to propose more variety in the exercises. Another perspective is the setting up of a randomized controlled study to establish the effectiveness of the new exercises on the level of physical activity, physical abilities, motivation and quality of life among nursing home residents.»
Statement: Past studies have confirmed the effectiveness of exercise training in nursing home residents: Provide a citation for this important statement
Effects of multicomponent exercise training intervention on hemodynamic and physical function in older residents of long-term care facilities: A multicenter randomized clinical controlled trial. Journal of Bodywork and Movement Therapies 28, 231-237.
https://doi.org/10.1016/j.jbmt.2021.07.009
- Pepera G, Krinta K, Mpea C, Antoniou V, Peristeropoulos A, Dimitriadis Z (2022). Randomized Controlled Trial of Group Exercise Intervention for Fall Risk factors Reduction in nursing home residents. Canadian Journal on Aging, 42 (1).https://doi.org/10.1017/S0714980822000265
- TrTraining Improves Strength, Balance, and Gait Speed in Elderly Nursing Home Residents. Naczk M, Marszalek S, Naczk A. Clin Interv Aging. 2020 Feb 7;15:177-184. doi: 10.2147/CIA.S234299. eCollection 2020.
Authors’ response : Thank you for these very interesting references. These articles have now been cited in our study.
Reviewer 2 Report
Congratulations to the authors for the theme under study. This report is well described and developed. However, some issues may need to be resolved. Some considerations are exposed.
It is suggested to insert keywords in the abstract of the report.
The introduction section presents a brief contextualisation of the normal human ageing process. However, it could also address the changes associated with ageing taking into account the importance of physical and social environments in influencing older people's physically active and sedentary opportunities, decisions and behaviours. Furthermore, despite the description of the health benefits of physical activity in the elderly, it would also be useful to briefly describe some of the negative health consequences of excessive mentally passive sedentary behaviour, complementing the idea of physical activity patterns and sedentary behaviours of the elderly. The aim of the study is well defined. However, it is not clear why, before the results are known, the development of 4 exercises was previously defined for the components discussed. Why, specifically, this number of exercises for each component? In another dimension, advanced age may also be related to the worsening of fine motor skills, and it is possible to include stimulation in physical activity programmes.
There is a high diversity of methods and techniques for qualitative data collection and analysis. Furthermore, certain techniques are linked to certain epistemological and ontological positions. In this sense, it would be useful to inform the readers about the philosophical assumptions that the researchers adopted in relation to the research object, concerning the epistemological position used (e.g., post-positivist, constructivist,...).
Despite the description of the participant selection process, there are some issues that need to be further clarified. In this study, no specific exclusion criteria were defined. However, defining a universe of a set of participants is not only a practical limit that assists the sampling process, but also possessed an important theoretical role in the process of analysis and interpretation. Therefore, it is suggested that some more criteria for the selection of participants could be included (e.g. the length of experience in relation to the application of physical exercises for the elderly, the academic background in that area, ...). Since as the data collection process occurs, there are reasons to change the size of the number of participants, within the parameters previously established. What was the procedure that allowed the decision to recruit 8 participants? Even though it is recommended to include 6 to 10 participants per focus group, there may be reasons, such as the power of information to properly guide the sample size. Information power indicates that the more diverse information the sample is able to produce, the smaller the number of participants becomes. In another sense, it may also make sense to classify the sampling strategy used.
It would be useful to indicate how long the focus group lasted. Were the topics designed on the basis of any previous research? Has their content been evaluated by researchers with experience in qualitative research? In addition, have the topics been the subject of a pilot study? This may shed some light on some of these aspects.
MURAL software was used to analyse data from the focus group. However, there is a marked diversity of data analysis techniques. It would be necessary to clarify the technique used in qualitative data analysis.
In addition, the use of procedures and techniques related to data credibility would also be useful (e.g. checking data quality across two, or more, researchers using Cohen's Kappa coefficient inter-judge agreement).
In order to highlight the results, it is suggested that some quotes be inserted in the words of the participants, complementing the latent analysis carried out.
There are some formatting flaws regarding the referencing system used in this journal. For example, "Waggoner et al. (2016)", page 11, line 3.
The authors state that it is difficult to compare the results of this study with other studies, because these do not specifically mention the exercises used in their physical activity programmes. However, it may be suggested that the exercises described here can be compared with other physical activity programmes for the elderly published in the literature.
Author Response
Congratulations to the authors for the theme under study. This report is well described and developed. However, some issues may need to be resolved. Some considerations are exposed.
It is suggested to insert keywords in the abstract of the report.
Authors’ response :The following key words have now been added to the manuscript :
« physical exercise; muscle strength; balance; flexibility; gait »
The introduction section presents a brief contextualisation of the normal human ageing process. However, it could also address the changes associated with ageing taking into account the importance of physical and social environments in influencing older people's physically active and sedentary opportunities, decisions and behaviours. Furthermore, despite the description of the health benefits of physical activity in the elderly, it would also be useful to briefly describe some of the negative health consequences of excessive mentally passive sedentary behaviour, complementing the idea of physical activity patterns and sedentary behaviours of the elderly. The aim of the study is well defined. However, it is not clear why, before the results are known, the development of 4 exercises was previously defined for the components discussed. Why, specifically, this number of exercises for each component? In another dimension, advanced age may also be related to the worsening of fine motor skills, and it is possible to include stimulation in physical activity programmes.
Authors’ response : As suggested by the reviewer, the introduction has been revised.
- The changes associated with aging that influence sedentarity have been discussed
- The negative heath consequences of sedentary behaviours have been added.
- The reason why 4 exercises in each category have been added has been explained
« Normal aging is accompanied by a deterioration in functional and locomotor capacity [1, 2], accentuated by physical inactivity and sedentary lifestyle which affects 50% of the elderly [3]. In addition, this deterioration decreases the mobility of the older adults, creating a vicious cycle of deconditioning [3], which accelerates the spiral of loss of autonomy, sarcopenia and increases the need for health care and services and therefore health costs [4, 5]. Unfortunately, inactivity and sedentary problems are even more prevalent in nursing homes. In fact, nursing homes residents spend the majority of their time inactive [6] and they walk on average 1678 ± 1621 steps per day, which is far from the recommendations levels advocating a minimum of 3000 steps/day [7, 8]. However, physical inactivity is the 4th risk factor for mortality [9]. In addition, lack of physical activity is detrimental to older adults’ health, functional independence and quality of life [3]. sedentary risk factors include intrinsic factors (e.g. physical health, attitudes related to aging, financial costs, lack of motivation, enjoyment, lack of companionship and knowledge of programs), extrinsic factors (e.g. transports, limited availability of physical activity pro-grams and lack of information on available activities, culture and sense of acceptance) and health- related factors (e.g. musculoskeletal disorders). On the other side, it is admitted that the implementation of physical activity interventions leads to positive effects on functional ability, cognition or mood in older adults [10-13]. In fact, being active is associated with body composition and functional capacities [14]. In addition, physical activity seems to be the to improve mitochondrial density and dynamics (i.e. resistance training) and id related to mitochondrial antioxidant capacity improvements (i.e. endurance training) [15]. Thus, Interventions should encourage oldest old adults to reduce sedentary time and es-pecially target mentally passive sedentary time [16].
This is why our team developed, a few years ago, a giant physical activity board game, the GAMotion, in order to promote physical activity in nursing homes. It is a medi-cal device of class 1. As shown in Figure 1, the GAMotion measures 3.5m long and 1.5 m wide and is composed of 12 squares, divided into 3 distinct colors corresponding to the 3 main components of physical activity: Balance (4 purple squares); muscle strength (4 yel-low squares); walking or endurance (4 red squares). In addition, the mat has 7 squares which represent the walking path, to perform the walking exercises (red squares). On each square, 3 levels of difficulty are represented by 1, 2 or 3 stars, so that the exercises are suit-able for the fitness levels of all participants. The principle of the game is similar to the tra-ditional goose game and the game only requires a dice and a chair.
Two previously published studies support the positive effects of the GAMotion on the level of physical activity and a broader array of physical and psychological outcomes [17, 18]. In the first publication, we showed that nursing home residents who used the GAMo-tion for 1-month period (3 times a week) increased significantly their daily number of steps and their daily energy expenditure but also their quality of life, balance, gait and strength of the ankle. More interestingly, these improvements still persisted 2 months after stopping the intervention with the GAMotion [18]. In the second publication, residents in-cluded in a 1-month intervention using the GAMotion displayed greater improvement in Tinetti score, Timed Up and Go test, Short Physical Performance Battery test (SPPB), knee extensor isometric strength, grip strength, symmetry of steps, 3 domains of the EQ-5D (i.e. mobility, self-care, usual activities) and intrinsic motivation, compared to the control group [17].
These results are promising but still have some limitations. First, a known barrier to the practice of physical activity among seniors is the lack of variety of exercises. Indeed, older adults, even in nursing home, need variety and innovation in their exercise pro-grams [19-21]. We should acknowledge that the GAMotion may seem monotonous due to the limited number of exercises offered. Second, the literature recommend flexibility train-ing, as supplement to other forms of exercises to improve functional ability of older adults [22]. To date, the flexibility category is not present in the GAMotion and this study will al-low us to add this category of exercise in a new version of the giant boardgame.
Thus, with a view to improving the current version of the GAMotion and countering the limitations mentioned above, the aim of this study was to develop 4 new balance, 4 flexibility, 4 muscle strength and 4 walking exercises to be included in the GAMotion (to replace the current exercises), to promote physical activity in nursing home settings.
Finally, the effects of the GAMotion on fine motor skills have not yet been studied but it would be really interesting to test this. In this sense, a future perspective has been added in the discussion section.
“Another perspective is to assess the effect of the GAMotion of fine motor skills of the nursing home residents »
There is a high diversity of methods and techniques for qualitative data collection and analysis. Furthermore, certain techniques are linked to certain epistemological and ontological positions. In this sense, it would be useful to inform the readers about the philosophical assumptions that the researchers adopted in relation to the research object, concerning the epistemological position used (e.g., post-positivist, constructivist,...).
Authors’ response : Thank you for raising this question. Our analysis based on positivism since we valued objectivity and proving or disproving hypotheses. Our epistemological position has been now described in the methods section.
« A two-steps qualitative study combining Focus group and Delphi method was conducted among healthcare professionals working in nursing home or having contact with nursing home residents. The philosophical research paradigms used to guide our qualitative methods is positivism because our methods resulted from foundationalism and empiricism. In fact, we valued objectivity and proving or disproving hypotheses [22] ».
Despite the description of the participant selection process, there are some issues that need to be further clarified. In this study, no specific exclusion criteria were defined. However, defining a universe of a set of participants is not only a practical limit that assists the sampling process, but also possessed an important theoretical role in the process of analysis and interpretation. Therefore, it is suggested that some more criteria for the selection of participants could be included (e.g. the length of experience in relation to the application of physical exercises for the elderly, the academic background in that area, ...). Since as the data collection process occurs, there are reasons to change the size of the number of participants, within the parameters previously established. What was the procedure that allowed the decision to recruit 8 participants? Even though it is recommended to include 6 to 10 participants per focus group, there may be reasons, such as the power of information to properly guide the sample size. Information power indicates that the more diverse information the sample is able to produce, the smaller the number of participants becomes. In another sense, it may also make sense to classify the sampling strategy used.
Authors’ response : we fully agree that more criteria for the selection of participants could have been included (e.g. the length of experience in relation to the application of physical exercises for the elderly, the academic background in that area, ...). Unfortunately, these information were not collected. Since these data could have influenced the results of the study, we added a limitation regarding the lack of these information
« The strength of this study lies in the rigorous methodology used to develop and validate the exercises but we must recognize that our study has some limitations. First of all, there is representativeness bias because not all professional categories related to nursing homes are represented. However, the professions most related to the GAMotion were included. Then, a recruitment bias is present since only volunteers were included via social media and emails. A information bias is also present since data regarding the length of experience in relation to the application of physical exercises for the elderly, the academic background in that area have not been collected. In addition, there was a significant dropout rate between the two rounds of the Delphi Method (round 1 n = 24, round 2 n = 15). Unfortunately, the reasons for abandonment are not known. This non-adhesion rate can be attributed to the relatively short expected time to respond (two weeks) or to the period in which it was launched (end of the academic year). »
It would be useful to indicate how long the focus group lasted. Were the topics designed on the basis of any previous research? Has their content been evaluated by researchers with experience in qualitative research? In addition, have the topics been the subject of a pilot study? This may shed some light on some of these aspects.
Authors’ response : These are indeed important elements. The duration of the focus group has now been specified (2 hours). The content of the discussion has been previously validated by an expert in the field of qualitative method and it has also been specified in the manuscript. However, a pilot study was not performed as part of this study.
« The first sample was asked to develop new exercises. To do this, a focus group lasting 2 hours was organized by videoconference, via Teams media, on May 25th 2021. The focus group was conducted in several steps, validated by an expert in qualitative method: 1) welcome and introduction of each participant, 2) presentation of the objectives of the discussion, 3) presentation of the GAMotion boardgame, 4) discussion about the difficulties encountered by the residents, 5) development of exercise ideas for each physical activity categories: balance, flexibility, muscle strength and walking exercises, 6) questions / answers session, 7) conclusion. During the steps 4 and 5 of the focus group, the different ideas emanating from the participants were noted on a virtual board using MURAL© software. That way, participants could observe this virtual board via screen sharing. The ideas were organized into diagrams according to each exercise category. ».
In addition, the use of procedures and techniques related to data credibility would also be useful (e.g. checking data quality across two, or more, researchers using Cohen's Kappa coefficient inter-judge agreement).
Authors’ response : only one researcher performed the statistical analysis. We admit that this is a limitation of the study. Therefore, this limitation has been added in the discussion section.
« The strength of this study lies in the rigorous methodology used to develop and validate the exercises but we must recognize that our study has some limitations. First of all, there is representativeness bias because not all professional categories related to nursing homes are represented. However, the professions most related to the GAMotion were included. Then, a recruitment bias is present since only volunteers were included via social media and emails. A information bias is also present since data regarding the length of experience in relation to the application of physical exercises for the elderly, the academic background in that area have not been collected. In addition, there was a significant dropout rate between the two rounds of the Delphi Method (round 1 n = 24, round 2 n = 15). Unfortunately, the reasons for abandonment are not known. This non-adhesion rate can be attributed to the relatively short expected time to respond (two weeks) or to the period in which it was launched (end of the academic year). Finally, a limitation concerns the statistical analysis because only one researcher performed the analysis and therefore the credibility of the results has not been tested (e.g. kappa-Cohen inter-judges) ».
In order to highlight the results, it is suggested that some quotes be inserted in the words of the participants, complementing the latent analysis carried out.
Authors’ response : Thank you for this suggestion. Some examples of comments received by the respondents have been added in the manuscript.
« Respondents commented on the exercises and these remarks mainly concerned the effectiveness of the exercises, the safety, the hygiene, the adequacy or the complexity for nursing home residents, and the possible variants. For example, a walking exercise on the heels, raising the point of the feet, was proposed. A comment concerning this exercise was "probably impossible because already complex for able-bodied people". Another example of comment concerns the stride of the dice placed on the ground: “it is a useful but complicated exercise for some residents who have difficulty bending their knees and lifting their feet off the ground". Still as an example, a proposed exercise consisted in joining the hands and raising them as high as possible. We received the following comment “it is an exercise often used in nursing homes and very popular with residents”. »
There are some formatting flaws regarding the referencing system used in this journal. For example, "Waggoner et al. (2016)", page 11, line 3.
Authors’ response : Thank you for highlighted this flag. The correction has been made.
The authors state that it is difficult to compare the results of this study with other studies, because these do not specifically mention the exercises used in their physical activity programmes. However, it may be suggested that the exercises described here can be compared with other physical activity programmes for the elderly published in the literature.
Authors’ response : We fully agree with the comment and we have compared our validated exercises with those used in published physical activity program.
« Thanks to the results of this research, future perspective is the creation of a new version of the GAMotion, using the new developed and validated exercises, in order to propose more variety in the exercises. Another perspective is the setting up of a randomized controlled study to establish the effectiveness of the new exercises on the level of physical activity, physical abilities, motivation and quality of life among nursing home residents. Another perspective is to assess the effect of the GAMotion of fine motor skills of the nursing home residents. For now, it is difficult to compare the results of this study with those found in the scientific literature because studies do not specifically mention the exercises used in physical activity programs intended for the older adults. However, we proposed a multi component physical activity program and numerous studies performed in nursing home setting recommend to implement combined physical activity intervention. In this sense, a me-ta-analysis including 12 studies, concluded that a combined PA program [type: strength and balance; frequency: 2 to 3 times/week; duration: 6 months and more] would prevent falls in nursing home residents with reduced mobility [37] Moreover, a systematic review showed that practicing combined moderate-intensity PA is the best intervention to improve quality of life, autonomy, balance, and anxiety in frail older adults [38]. »